# Insight Study on the Comparison between Zinc Oxide Nanoparticles and Its Bulk Impact on Reproductive Performance, Antioxidant Levels, Gene Expression, and Histopathology of Testes in Male Rats

**DOI:** 10.3390/antiox10010041

**Published:** 2020-12-31

**Authors:** Amira A. Goma, Hossam G. Tohamy, Sara E. El-Kazaz, Mohamed M. Soliman, Mustafa Shukry, Ahmed M. Elgazzar, Rashed R. Rashed

**Affiliations:** 1Department of Animal Husbandry and Animal Wealth Development, Faculty of Veterinary Medicine, Alexandria University, 22785 Alexandria, Egypt; amira.gomaa@alexu.edu.eg (A.A.G.); saraelkazaz@alexu.edu.eg (S.E.E.-K.); rashed.ragab@alexu.edu.eg (R.R.R.); 2Department of Pathology, Faculty of Veterinary Medicine, Alexandria University, 22785 Alexandria, Egypt; hossam.gafaar@alexu.edu.eg; 3Clinical Laboratory Sciences Department, Turabah University College, Taif University, P.O. Box 11099, Taif 21944, Saudi Arabia; 4Biochemistry Department, Faculty of Veterinary Medicine, Benha University, 13737 Benha, Egypt; 5Department of Physiology, Faculty of Veterinary Medicine, Kafrelsheikh University, 33511 Kafrelsheikh, Egypt; 6Department of Forensic Medicine and Toxicology, Faculty of Veterinary Medicine, Alexandria University, 22785 Alexandria, Egypt; elgazzar@alexu.edu.eg

**Keywords:** zinc oxide nanoparticle, testes, sexual behavior, antioxidant, bulk, semen

## Abstract

*Background:* Despite the beneficial effects of zinc oxide nanoparticles (ZnONPs) on different biomedical applications, including their antioxidant and anti-inflammatory ones, it might have cytotoxic and genotoxic impacts on the male reproductive system. *Objective:* The current study compares the effect of zinc oxide nanoparticles and their bulk form, at different doses, on male rats’ reproductive performance, testicular antioxidants, gene expression, and histopathology. *Materials and Methods:* Thirty male rats were randomly allocated equally in five groups. The control one was injected with Tween 80 (10%). The zinc oxide nanoparticle (ZnONP) groups received ZnONPs < 50 nm, specifically, 5 mg/kg (ZnONP-1) and 10 mg/kg (ZnONP-2). The bulk zinc oxide (BZnO) groups were administered 5 mg/kg (BZnO-1) and 10 mg/kg (BZnO-2), correspondingly. Rats were injected intraperitoneally with the respected materials, twice/week for eight consecutive weeks. Finally, the male rats’ sexual behavior and their pup’s performance were determined in a monogamous mating system. Rats were then anesthetized and sacrificed for semen characteristics evaluation and tissue collection for antioxidant and hormones analysis, gene expression, and histopathological examination. *Results:* It was shown that ZnONP-1 improved sexual behavior, semen characteristics, and pup’s performance compared to its bulk form. Similarly, the testicular antioxidants activity, glutathione (GSH), and superoxide dismutase (SOD) increased with a decrease in the malonaldehyde (MDA), interleukin 6 (IL-6), and tumor necrosis factor (TNF-α) levels. It also improves the reproductive hormone levels and mRNA expression of different steroidogenesis-associated genes and anti-apoptotic genes. *Conclusion:* It can be concluded that zinc oxide nanoparticles, administered at 5 mg/kg, had the most beneficial effect on male reproductive performance, while 10 mg/kg could have a detrimental effect.

## 1. Introduction

Nanotechnology is the science that deals with producing materials in nano-dimensions through the re-arrangement and re-location of its atoms to get materials with potentially better functions and properties. The use of nanoparticles is not only limited to industry and medicine but is also increasingly used in cosmetics, sunscreens, and food products [1]. The ability of nanoparticles to induce toxicity is mainly attributed to the increased surface reactivity of the materials [2]. The small size of nanoparticles shows a wide surface area per unit mass and make it more reactive inside the cell. Moreover, it has been confirmed that an increase in the nanoparticles surface area greatly increased the cell’s ability to generate reactive oxygen species [3], which harm the body and organs’ activity. 

The traditional zinc oxide form had been used as a feed additive for animal feeds more than other salts because it contains a greater zinc percentage and has a less toxic effect [4]. The zinc oxide nano form had been found to have special physical and chemical properties, even though it could throw out some of the body’s barriers for effective targeting of cells for its beneficial effect [5]. The existence of a high zinc concentration in seminal fluid could negatively affect the reproductive performance of male rats [6], as was informed by [7], who found that the intranasal irrigation with zinc sulphate completely abolishes their sexual behavior; indeed, [8] also indicated that excessive zinc intake in mice adversely affects sperm count and motility. 

The in vitro toxicity of zinc oxide nanoparticles (ZnONPs) has been reported [9]. Furthermore, the developmental toxicity of ZnONPs on the fetus also has been shown in some papers [10]. Still, preliminary studies explain its internal mechanism, as Chen et al. [11] found that exposure to ZnONPs in high doses reduced the fetal number and their weight. Moreover, Teng et al. [12] showed that ZnONPs affect placental weight, rat weight, and vitality. The current study investigated the zinc oxide nanoparticle effect compared to its bulk form with different doses on male reproductive performance, testicular antioxidants, reproductive hormones, gene expression, and histopathology. 

## 2. Materials and Methods

### 2.1. Chemicals

The ZnONPs < 50 nm were obtained from Sigma-Aldrich Co., St. Louis, MO, USA, bulk zinc oxide (BZnO) from Oxford Lab Chem Co., Mumbai, India, and Tween 80 obtained from Alpha Chemika Co., Mumbai, India.

### 2.2. Characterization of ZnONP

Transmission electron microscopy (TEM) was used to determine the ZnONPs’ morphological characteristics at the University of Alexandria, Egypt. The NPs were dispersed in distilled water and ultrasonicated for about 15 min. Then we deposited the dispersion on copper grids, dried them, and set them up for TEM. The diameter of each particle was evaluated [13].

### 2.3. Animals and Experimental Procedures

Thirty male Sprague-Dawley rats (120–130 g) were supplied by the Medical Research Institute, Alexandria University, Egypt. The Institutional Animal Care and Use Committee have authorized the experimental procedures at Kafrelsheikh University, Egypt (Number-2018-/1155). After one week of adaptation, rats were randomly and equally assigned into five groups (*n* = 6). The control group was injected with Tween 80 (10%). The ZnONP groups received a ZnONP dose of 5 mg/kg (ZnONP-1) and 10 mg/kg (ZnONP-2). The bulk zinc oxide (BZnO) groups were administrated BZnO of 5 mg/kg (BZnO-1) and 10 mg/kg (BZnO-2), based on a previous study [14]. All drugs were dispersed in Tween 80 (10%), and each rat was injected intraperitoneally according to its weight to reach the desired dose, twice/week for eight consecutive weeks. The diet provided met the NRC recommendations [15]. At the end of the treatment period, the rats were first checked for sexual performance, fertility, and decapitated for sampling. Every group was checked blindly at both stages concerning the other treated groups. Tests were compiled and divided into two equal sections. The first part was kept at −80 °C for further antioxidant markers investigation, reproductive hormones analysis, and gene expression, whereas for histopathological examination, the second portion was submerged into formalin solution.

### 2.4. Sexual Behavior (Fertility Test)

Thirty female rats were injected intraperitoneally with *Lutalyse*^®^ (dinoprost tromethamine) at 0.1 mg/100 g bwt [16] twice/day, in synchronization with estrous. A vaginal smear was performed to check the female in estrous. Estrous females were then mated in a monogamous mating system (1 male:1 female) in a plastic cage and video-recorded for 15 min. The mount and intromission numbers, in addition to mount, intromission, and ejaculatory latency, were recorded [17]. Again a vaginal smear was performed to detect sperm, and the sperm-finding day was set as day zero of gestation [18].

### 2.5. Pup’s Performance

The following parameters were determined for the pups delivered, as discussed earlier [19]: Total litter size (number): total pups born; live litter size (number): a live-born pups; and dead litter size: dead pups number at birth. Birth weight: pup weight per dam at birth; and weaning weight: pup weight per dam at weaning. Survival rate: percentage of the maintained live pups from the total number of pups born per dam; and conception rate: percentage of pregnant females per group.

### 2.6. Semen Characteristics

The semen analysis was performed [20] as follows:Sperm motility: By cutting the cauda epididymis in a sterile petri dish, sperm were collected to allow the sperm to bathe out of the epididymal tubules. There was a drop of sperm suspension on a slide and a coverslip on it. Ten fields were examined at ×400 magnification by a phase-contrast microscope. The % of motile sperm from the total sperms counted was evaluated within 2−4 min.Sperm viability: A sperm suspension drop was mixed with one drop of 1% eosin Y/5% nigrosine, and a smear was made. Upon 2 min of incubation at room temperature, the slides were examined with magnification by a bright-field microscope at ×400. Per sample we counted one hundred sperms and the % viability was measured. Dead sperm were pink and live sperm were not stained.Sperm abnormalities: A drop of 1% eosin Y/5% nigrosine was put into one drop of the sperm suspension. Sperm smears were pulled out on clean, grate-free slides, and a hundred sperm were counted at ×400 for morphological abnormalities, such as amorphous, bicephalic, spiral, or irregular tails.Epididymal sperm count: Five μL of epididymal sperm suspension was diluted by 95 μL of solution (5 g NaCl and five drops of formalin/100 mL distilled water). One drop of the diluted epididymal content was moved to each of the hemocytometer counting chambers and allowed to dry for 5 min. During this period, the cells were deposited and numbered at ×400.

### 2.7. Reproductive Hormones Assay in Testes 

Rats were anesthetized by sodium pentobarbital (60 mg/kg, Sigma-Aldrich Co., St. Louis, MO, USA), after which the testes were collected. Briefly, 100 mg of tissue was washed with 1× PBS (phosphate-buffered saline), then homogenized in 1 mL (1× PBS), and kept overnight at −20 °C. The homogenates were thawed and centrifuged at 5000× *g*, at 4 °C for five minutes. The supernatants were immediately collected and analyzed. The protein concentration for the testicular homogenates was assessed by Bradford assay for normalization of the biochemical analysis. The testosterone level was assessed as before [21], using ELISA kits (Immunometrics Ltd., London, UK). The follicle-stimulating hormone (FSH) and luteinizing hormone (LH) was also ascertained using ELISA kits (Catalog Number: FSH (# MBS2502190) and LH (# MBS161787), My BioSource, San Diego, CA, USA respectively).

### 2.8. Testicular Oxidative Markers

After homogenization in PBS, testicular tissues were primed for oxidative markers assessment, and the supernatants were used for MDA [22] and GSH [23] concentrations and SOD [24], utilizing commercially available kits (Bio-diagnostic, Giza, Egypt). The IL-6 and TNF-α concentrations were assessed using an ELISA kit (Catalog Number: MBS355410 and MBS282960, respectively, My BioSource, Inc., San Diego, CA, USA). For standardization of the biochemical parameters, the protein concentrations were evaluated through the Bradford test.

### 2.9. Real-Time Polymerase Chain Reaction (RT-PCR)

Quantitative RT-PCR was used to determine the relative expression of testicular genes. In brief, TRIZOL Reagents (Invitrogen, life science, Carlsbad, CA, USA) and Nanodrop used for quantification were extracted for the total RNA of approximately 100 mg testicular tissue. For the DNA synthesis, using cDNA synthesis (Fermentas, Waltham, MA, USA), samples of 1.8 or more A260/A280 RNA were used. The SYBR Green Master Mix and primers in Table 1 were applied to improve the cDNA, utilizing the GAPDH household gene. Data on amplification were analyzed using 2^−ΔΔT^ methods [25].

### 2.10. Histopathology

Testicular tissue samples were soaked and fixed in 10% neutral buffered formalin (NBF) for at least one day. Fixed specimens were obtained using the traditional paraffin-embedding technique. Sections of 5 µm slices were stained with Mayer’s hematoxylin and eosin (HE). The incidence and severity of the histological lesions were determined. Incidence is the number of rats with lesions per total examined (6 rats per group). The severity of the lesions was graded by estimating the percentage of area affected in the entire section. Lesion scoring: (-) absence of the lesion = 0%; (+) mild = 5–25%; (++) moderate = 26–50%; and (+++) severe = 50% of the examined tissue sections. Assessment of the spermatogenesis indexes was done using Johnsen scores to sort spermatogenesis at 400× magnification power. To evaluate the specimen’s quality, we used a slightly modified Johnsen score count [26]. Microscopically, the spermatogenesis values were classified as 1, for Sertoli cells only; 2, for spermatogonia only; 3 and 4, for no further than primary spermatocytes (40 spermatids per view, respectively); 8, 9, and 10, for maturation-phase spermatids (50 spermatids per view, respectively).

### 2.11. Immunohistochemistry and Quantitative Analysis

The ki-67 expression used to detect the proliferative index was determined utilizing the streptavidin–biotin complex immune-peroxidase system. Paraffin-embedded parts were dewaxed and incubated in 0.1% H_2_O_2_ block endogenous peroxidase for half an hour, and then rinsed three times in PBS. Antigen-retrieval by microwave treatment was as follows: 20 min, 0.01 Mol/L citrate buffer, and pH 6. The slides were incubated overnight with the primary antibody. For detection of Ki-67, sections were for Ki-67 recognition; sections were incubated with rabbit monoclonal Ki-67 antibody (SP6) diluted at 1:200 and added to sections at 40 °C for 12 h. After several PBS washes, the primary antibodies were identified by incubation with biotinylated anti-rabbit antibodies for 30 min at room temperature. The sections were then incubated for 30 min at room temperature with the streptavidin–biotin peroxidase complex. For the next wash-up with PBS, the responses were conceived with 3, 3′-diaminobenzidine-tetrahydrochloride chromogen to envision the antibody attachment. The sections were counterstained with Mayer’s hematoxylin, dehydrated, and mounted. For the negative control, the primary antibody was substituted with PBS. The positive reaction for Ki-67 was nuclear brown particles in the cytoplasm. Images of 10 different fields at ×400 were examined using ImageJ software to assess the area % of Ki-67-positive brown immunostaining cells.

### 2.12. Data Analysis

Statistical analyses were performed via one-way GLM variance analysis using SAS (Statistical Analysis Framework, Version 6, 4th Edition, SAS Institute, Cary, NC, USA). P values <0.05 were considered significant unless otherwise specified. The Duncan multi-range test was used to evaluate the major effects of the experimental treatment.

## 3. Results

### 3.1. Characterization of the ZnONPs

Transmission electron microscopy (TEM) revealed that the diameter of the ZnONPs was <50 nm, as shown in Figure 1.

### 3.2. Sexual Behavior

Figure 2A revealed a substantial (*p* < 0.0001) rise in the ZnONP-1 group for mount and intromission numbers than the other treated groups; however, there was a decrement in the high dose groups of both forms, with the least in the bulk one. Although there was a non-significant difference between BZnO-1 and the control group, the ZnONP-2 group showed a decrease compared to the control. In Figure 2B (*p* < 0.0001), latencies to mount and intromission were the longest in the BZnO-2 group, whereas the shortest in the ZnONP-1 group compared to the other treated groups. In contrast, there was a non-significant variation between BZnO-1 and the control group, although, the ZnONP-2 group was longer than the control. In Figure 2C, the ejaculatory latency (*p* < 0.0001) was the shortest in the ZnONP-1 group; however, the BZnO-2 group was the longest compared to the other groups. There was also a substantial rise in the ZnONP-2 and BZnO-1 groups when compared to the control.

### 3.3. Semen Characteristics

Figure 3A,B revealed that the ZnONP-1 group was the highest (*p* < 0.0001) in sperm motility, viability, and count, while the BZnO-2 group was the lowest compared to the other treated groups. Although there was no substantial difference between the BZnO-1 and control groups, the ZnONP-2 group displayed a decline in sperm motility and percentage viability from the controls. Simultaneously, sperm count was higher in the BZnO-1 group than the control, while lower in the ZnONP-2 group. Figure 3C show that the highest sperm abnormalities percentage (*p* < 0.0001) was in the BZnO-2 group, whereas the least was in the ZnONP-1 group. A non-meaningful disparity was discovered between BZnO-1 and the control; however, the ZnONP-2 group was higher than the control. Regarding spermatogenesis, the Johnsen scores were 9.01, 9.51, 5.61, 8.11, and 4.51 for the control, ZnONP-1, ZnONP-2, BZnO-1, and BZnO-2 groups, respectively (Figure 3D).

### 3.4. Pup’s Performance

From the results in Figure 4D we deducted that the conception rate (%) decreased (*p* = 0.02) in the high dose groups for both forms, although there was a difference between them in the pup’s performance. Total and live litter numbers in Figure 4A increased (*p* < 0.0001) in the ZnONP-1 group while decreasing in the BZnO-2 group compared to the other treated groups. However, a decrease in the ZnONP-2 group compared to the control was contrasted by an increase in the BZnO-1 group. Dead litters (*p* < 0.0001) were present only in the high dose groups of both forms. Figure 4C shows that the survival rate (%) decreased (*p* < 0.0001) in the BZnO-2 group and ZnONP-2 when compared to the other treated groups, with the least so in the BZnO-2 group. Birth and weaning weights (g/dam) in Figure 4B show an increment (*p* < 0.0001) in ZnONP-1 pups; however, there was a decrement in the BZnO-2 pups. A non-substantial disparity was found between BZnO-1 and the control group for birth and weaning weight, although, the ZnONP-2 group was lower than the control.

### 3.5. Reproductive Hormones

As shown in Figure 5, there was a noteworthy increase in the testosterone, FSH, and LH hormones level in the ZnONP-1 group compared to the other treated groups, with no substantial difference from the control group; however, there was a decrement in the high dose groups from both forms, the least so in the BZnO-2 group. In addition, there was a significant improvement in the BZnO-1 group compared to the BZnO-2 group.

### 3.6. Testicular Oxidative Markers

Results in Figure 6A,C show that the GSH and SOD levels significantly increased in the ZnONP-1-treated group compared to the other treated groups. There is a significant increase in the BZnO-1 group compared to the BZnO-2- and ZnONP-2-treated groups in the same context. Figure 6B shows that the testicular level of MDA was significantly decreased in the ZnONP-1 group compared to the other treated groups; nevertheless, the BZnO-2 group was higher. In turn, the BZnO-1-treated group showed a significant decrease compared to the ZnONP-2 and control groups. In Figure 6D,E there was a significant decrease in the IL-6 and TNF-α levels in the ZnONP-1 group compared to the other treated groups, especially the ZnONP-2- and BZnO-2-treated groups; this besides the substantial decrease in the BZnO-1 group relative to the treated groups ZnONP-2 and BZnO-2.

### 3.7. Gene Expression Findings

In Figure 7A–G,I the gene expression analysis revealed that there was a significant up-regulation in the mRNA expression of CYP17A1, cytochrome P450 17A1, STAR, Cyp11a1, P450SCC, 3β-HSD, CYP19, and cytochrome P450 aromatase, as well as for the androgen receptors (AR), luteinizing hormone receptors (LHR), and Bcl-2- in the ZnONP-1-treated group in comparison to the other treated groups. In addition, there was a significant up-regulation in the BZnO-1 group in comparison to the other treated groups. In the same context, Figure 7H indicates a significant downregulation of Bax in both the ZnONP-1 and BZnO-1 groups compared to the BZnO-2 and ZnONP-2 groups, respectively. 

### 3.8. Histopathological Findings

The histopathology of the testes was evaluated under a light microscope. Testes of the control mature rats showed uniform seminiferous tubules with full spermatogenesis and interstitial tissue (Figure 8a), while testicular sections of the ZnONP-1 rats revealed a normal morphology (Figure 8b). However, testes of the ZnONP-2 rats showed deteriorating variations that were portrayed by small, shrunken, buckled, unsystematic seminiferous tubules (Figure 8c); this besides a reduction in germinal cells, hyalinization of the luminal contents, with vacuolar degeneration of the germinal epithelium and Sertoli cells (Figure 8d), germinal epithelium sloughing in the seminiferous tubules (Figure 8e), as well as interstitial edema, defined by the faint albuminous eosinophilic content (Figure 8f).

On the other hand, testicular sections of the BZnO-1 rats revealed a normal morphology (Figure 9a). In contrast, testes of the BZnO-2 rats exhibited a multi-nucleated giant cell in the lumen of some seminiferous tubules (Figure 9b), with fragmentation and necrosis of the germinal epithelium (Figure 9c); this besides necrotic seminiferous tubules, a reduction in germinal cells, and hyalinization of the luminal contents (Figure 9d), sloughing of the germinal epithelium in the lumen of the seminiferous tubules (Figure 9e), and interstitial edema with mild inflammatory cell infiltration (Figure 9f). The prevalence and severity of the lesions in the ZnONP-2- and BZnO-2-treated groups are summarized in Table 2.

### 3.9. Immunohistochemistry and Quantitative Analysis

Testicular sections of the control and ZnONP-1 rats stained with anti-Ki-67 antibody showed a stronger positive immunoreaction in the nuclei of the spermatogonia and primary spermatocytes (Figure 10a,b). In contrast, the testicular sections of the ZnONP-2 rats presented a positive immunoreaction in the nuclei of some spermatogonia and primary spermatocytes and negative immunoreaction in other germ cells (Figure 10c). On another side, the testicular sections of the BZnO-1 rats exhibited a powerful positive immunoreaction in the spermatogonia and primary spermatocytes (Figure 10d). However, the BZnO-2 rats revealed a negative immunoreaction in most nuclei of the spermatogonia and primary spermatocytes (Figure 10e). The mean area percent of Ki-67 immunostaining positive cells was significantly decreased in the ZnONP-1 and BZnO-1 rats, whereas a marked significant decrease was present in the ZnONP-2 and BZnO-2 rats in comparison to the control rats (Figure 10f).

## 4. Discussion

Not all levels of nanoparticles (NPs) have a protective or non-toxic impact on spermatogenesis [27], compared to its traditional forms. Recent studies deducted that some NP levels have a toxic effect on male germ cells [28] through the creation of reactive oxygen species (ROS), resulting in oxidative damage [29]. Therefore, the present study’s aim was to determine the difference between the effect of ZnONPs and its bulk form in two doses on male reproductive performance, testicular antioxidants, reproductive hormones, gene expression, and histopathology. This study hypothesized that the ZnONPs could positively affect the male rat reproductive system compared to its traditional form but at a certain level as its higher level could hurt it.

The current study showed that ZnONP-1 had improved male sexual performance. However, no difference was found between BZnO-1 and the control for all parameters, except that the ejaculatory latency and sperm count were higher in BZnO-1 than in the control. In contrast, ZnONP-2 and BZnO-2 hurt sexual performance, with a more adverse effect for BZnO-2. This could be attributed to the NPs’ administration at low doses, which might have a defensive impact through their antioxidant processes, while high doses could have a harmful effects [30]. This was supported by [31], who reported that ZnONP at a low dose (5 mg/kg) could have a beneficial effects on male reproduction. However, high doses (50, 150, 300, and 350 mg/kg) could act as testicular toxicants in mice. The authors in [32] also revealed that ZnONP treatment in mice at low doses is relatively biocompatible as a nutritional additive after long-term feeding when compared to their microparticles. However, [33] revealed that ZnONPs (10 mg/kg bwt) increased sperm quantity and motility in diabetic rats. On the other hand, [7] reported that the intranasal irrigation with zinc sulfate solution of male mice leads to abolishing sexual behavior in the mating test with an estrous female. 

Normal sperm functioning requires a low level of reactive oxygen species (ROS) physiologically, and if the ROS levels are above the physiological level, sperm functioning deteriorates [34]. ROS and their metabolites are capable of targeting DNA, proteins, lipids, altering enzyme systems, and inducing cell death, and ultimately reducing the semen parameters associated with male infertility. [35]. Reference [36] deducted that the release of metallic cations Zn2+ could be the main cause of toxicity. That explained what was reported in the present study in the BZnO-2 and ZnONP-2 groups, in which the semen characteristics of the male rats were adversely affected, which was more pronounced in BZnO-2. It has been shown that an abnormal sperm morphology depends entirely on the spermatogenesis phases regulated by Sertoli cells [37]. Additionally, the reduction in the number of sperms is associated with reducing the number of spermatids and spermatogonia cells [38]. That might be a corollary of the apoptotic effects that high-level ZnONPs have on spermatogenic cells [39]. This was reported by [40], who deducted a reduction in sperm motility percentage and an upsurge in the abnormal sperm percentage in rats treated by ZnONPs at 700mg/kg; also, the sperm abnormalities percentage was significantly increased with the administration of ZnONPs at 100 and 200 mg/kg for 7 and 14 days [41]. The spermatic abnormalities reached 75% at 100 mg/kg and 95% at 200 mg/kg, which showed the stronger effect of a high ZnONP level on sperm structure. 

This was inconvenient with [42], who showed that 5 mg/kg of ZnONPs resulted in testicular necrosis and male rats’ degeneration. Reference [39] demonstrated that ZnONP 5 mg/kg treated mice daily for 35 days revealed no variation in testicular markers, including sperm quantity, sperm motility, and sperm deformity, when associated with the control. No obvious acute toxicity was detected in mice treated with intravenous injection of 30 mg/kg ZnONPs for two weeks [43]. Reference [44] mentioned that this variation in nanoparticle injuriousness depends on the size, dose, administration route, and duration of exposure. Therefore, adequate nano zinc consumption is essential for preserving the ideal testicular function [45].

There is a lot of evidence showing that fetuses are more affected by nanoparticles than adults because of their physiological immaturity [46]. In what [47,48] have described, nanomaterials could diffuse the blood–placental barrier, explaining its beneficial effect on pup’s performance in the present study for ZnONP-1.

In contrast, ZnONP exposure during the organogenesis stage led to poor embryonic development in mice [49]. Jo et al. [50] also indicated that exposure to ZnONP before and during gestation could compromise pregnant females’ health and fetuses. This was reported in the present study in the BZnO-2 and ZnONP-2 groups. The conception rate was lowered with a decrement in total litter and live litter numbers and the presence of dead litter, and a lower survival rate for the live ones. In addition, there were decreased birth and weaning weights found in these pups, which was the least in BZnO-2. Chen et al., [11] supports these findings, in which they stated that the exposure to ZnONP in pregnant mice prompted dam injury, fetal growth limitation, and a decrease in the fetal numbers. They contributed to placental damage and functional alterations triggered by apoptosis, oxidative, and endoplasmic reticulum anxiety. Woods et al. [51] revealed that placental dysfunction could affect fetal growth limitation and miscarriage. Similarly, a decrease in fetal weights were seen after administration of ZnONPs at 400 mg/kg/day [52]. However, no effect was reported on their conception rate (%), litter size, fetal deaths, and weight [50]. Therefore, these particles have a particular concern for gonadal and reproductive toxicity and need further study.

Zinc has several significant biological hormone reactions [53]. Zinc had a function in hormone storage, synthesis, and discharge, to improve the passivity of organ reactivity and receptor positions [54]; this in addition to its critical role in many sex hormones, including testosterone [55], for which, ZnO could be one of the issues that lead to the elevation of testosterone concentrations when used as a resource of zinc in the industry [56]. The present results concerning the reproductive hormones revealed a notable rise in testosterone, FSH, and LH hormones level in the ZnONP-1 group relative to the other treated groups, with a significant improvement in BZnO-1 with the BZnO-2-treated groups; these results supported our Johnsen score results, which reflects improved spermatogenesis in the ZnONP-1 and BZnO-1 groups compared to the ZnONP-2 and BZnO-2 groups. These results are inconsistent with Mozaffari et al., [57], who stated that zinc could lead to a substantial rise in the testosterone hormone level and an increase in FSH hormone level in rats treated with ZnONPs. 

The FSH hormone level change might be due to nanoparticles moving out of the blood–brain barrier since FSH and LH are pituitary hormones [58]. There are two intracellular pathways by which ZnONPs improve the FSH and LH. The first is the increased pituitary zinc levels that cause destruction and decrease in the pituitary dopamine release and, therefore, decrease the inhibiting effect of dopamine on the GnRH, and this was an indirect effect. The second was the direct effect of ZnONPs in inhibiting the GABA nerve pathway [57].

Zinc is also a cofactor for more than 200 enzymes with various significant roles, mainly those involved in the antioxidant defense mechanism. The antioxidant and anti-inflammatory activity of ZnONP was prominent in this study in which GSH and SOD were significantly increased in their levels. However, MDA, IL-6, and TNF decreased in the ZnONP-1-treated group compared to the other treated groups. This was consistent with [59,60]. They reported that ZnONPs could minimize testicular tissue oxidative stress with decreased MDA levels and higher antioxidants (SOD, C.A.T., GPx, and GR) levels. This higher zinc concentration in the testicular tissue resulted from ZnONP dissociation, which had a potent antioxidant activity. Zinc is also the cornerstone of the antioxidant enzymes, such as SOD, and known as a sulfhydryl group protector, removing it from catalytic sites. Transitional metals, such as iron and copper, are also expected to damage lipid peroxide [61]. Therefore, ZnONP could protect the cell membrane’s integrity from exposure to the harm of oxidative stress, raising the levels of antioxidants, and minimizing the free radical levels [42]. Nagajyothi et al. [62] stated that ZnONP scavenged 45.46% DPPH at 1 mg/mL and showed an excellent dose-dependent suppression of both the mRNA and protein expressions of INOS, COX-2, IL-1β, IL-6, and TNF-α, which endorse the anti-inflammatory effect of the ZnONPs.

Zinc also stabilizes the nuclear chromatin of spermatozoa and its cell membrane in seminal fluids [63]. The obtained result of gene expression of steroidogenesis genes supports our hypothesis, in which there was a significant up-regulation in the mRNA expression of CYP17A1, cytochrome P450 17A1, STAR, Cyp11a1, (P450SCC), 3β-HSD, Cyp19, and cytochrome P450 aromatase; also in AR and LHR in the ZnONP-1-treated group relative to the other treated groups and a significant up-regulation in BZnO-1 compared to the BZnO-2-treated group. Komatsu et al., [64] indicated that nanoparticles might alter the expression of steroidogenic acute protein (STAR), which is accountable for cholesterol transport, which increases steroid hormone production [65]. Therefore, ZnONPs transport cholesterol into the inner mitochondrial membrane via increased expression of the STAR protein and then transform cholesterol into pregnenolone, which raises the testosterone concentrations. 

Different histopathological changes were found in the ZnONP-2 and BZnO-2 rats, such as shrunken, clasped, unsystematic seminiferous tubules, and vacuolation of tubular epithelial and Sertoli cells, which are the supportive cells within the seminiferous tubules, making up a multitude of factors required for spermatogenesis [65]. Likewise, some tubules showed hyalinization of the luminal content. Others had shed germinal epithelial cells within their lumina due to injured Sertoli cell because of microtubule damage [66]. These histopathological findings might be because of the lowered serum testosterone levels. Concerning the expression of Ki-67 antibody in these groups, the cell proliferation was decreased because of the high level of zinc’s inhibitory role on cell proliferation, which elucidates the dented seminiferous tubules in rats treated with elevated doses of ZnONPs and BZnO.

## 5. Conclusions

The present study was the first one investigating the effect of different forms of zinc salt, either as nanoparticles or bulk, on the male reproductive performance, hormones level, antioxidant levels, gene expression, and histopathology of the testes. It can be concluded that ZnONPs at 5 mg/kg improved the sexual behavior, semen characteristics, and pup’s performance, also increasing the testicular antioxidant activity, with significant improvements in reproductive hormones as well as mRNA expression of the different steroidogenesis and anti-apoptotic genes with the same dose of the bulk form. Finally, zinc oxide nanoparticles at 5 mg/kg showed the most beneficial effect on rat reproduction.

## Figures and Tables

**Figure 1 antioxidants-10-00041-f001:**
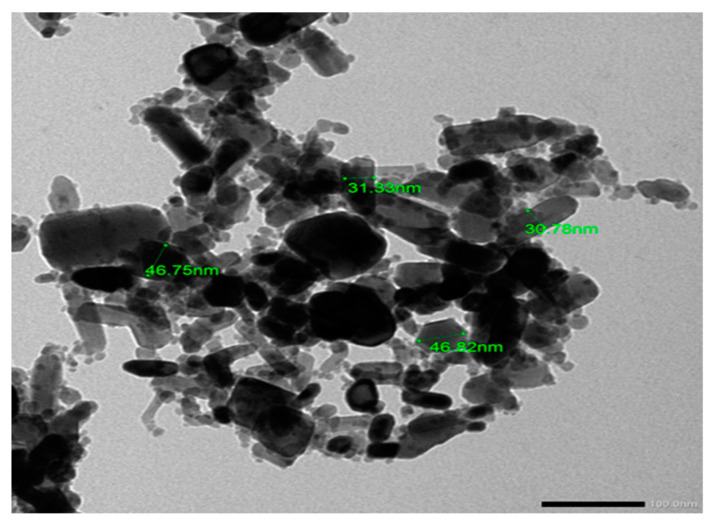
TEM image showing the shape and diameter of the zinc oxide nanoparticles (ZnONPs) being less than 50 nm (scale bar = 100 nm).

**Figure 2 antioxidants-10-00041-f002:**
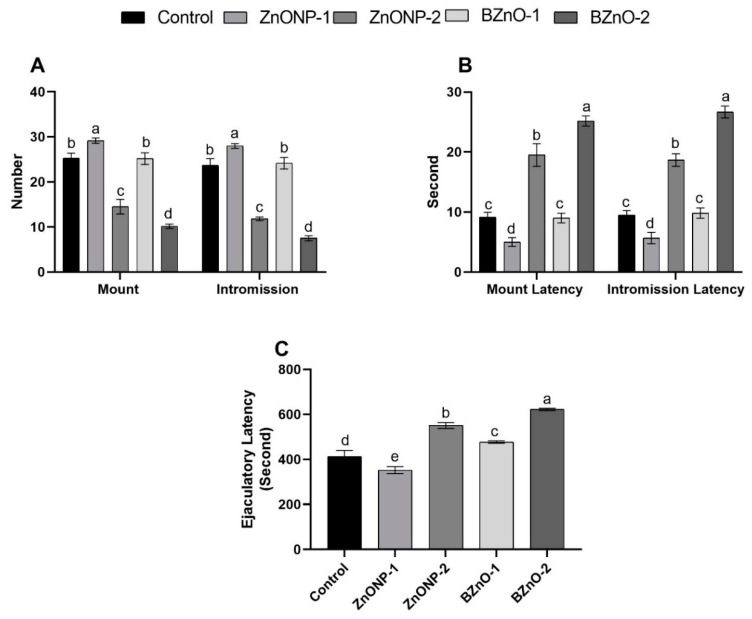
Effect of the ZnONPs and bulk zinc oxide (BZnO) at different doses on the sexual behavior of male rats: (**A**) mount and intromission numbers; (**B**) mount and intromission (seconds) latencies; and (**C**) ejaculatory latency (seconds). All the values are expressed as the mean ± SEM. Different small letters (a–e) indicate significance at *p* < 0.0001.

**Figure 3 antioxidants-10-00041-f003:**
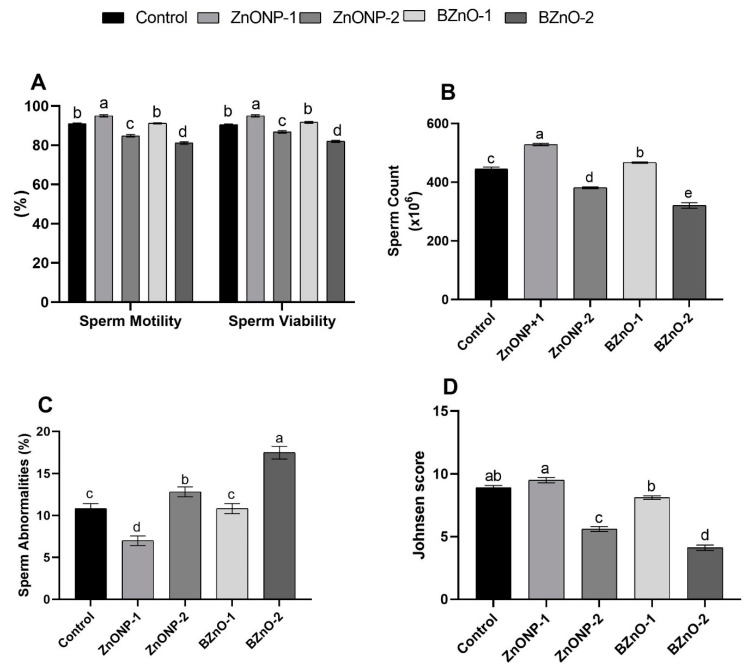
Effect of the ZnONPs and BZnO at different doses on the semen characteristics of male rats: (**A**) sperm motility (%) and viability; (**B**) sperm count (×10^6^); (**C**) sperm abnormalities (%); and (**D**) Johnsen score. All the values are expressed as the mean ± SEM. Different small letters (a–e) indicate significance at *p* < 0.0001.

**Figure 4 antioxidants-10-00041-f004:**
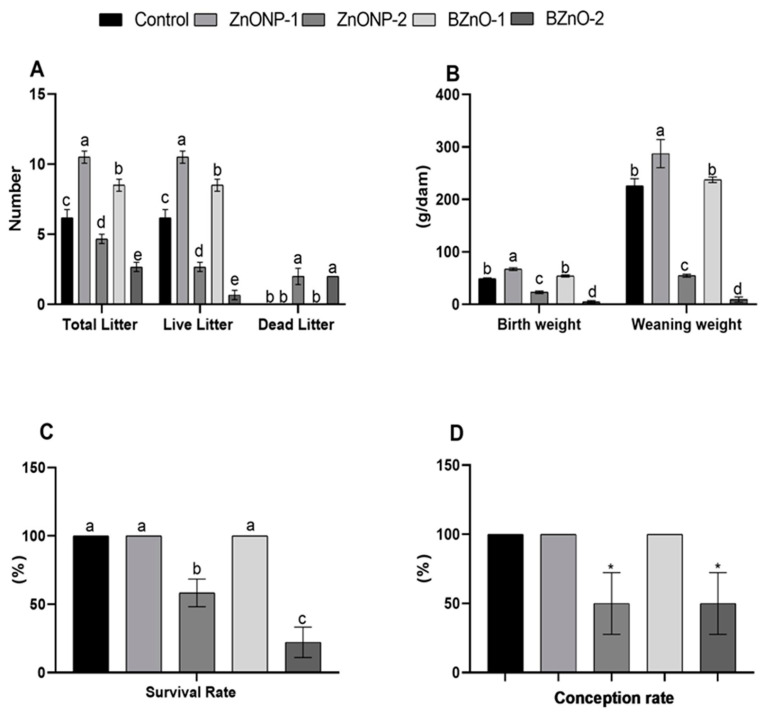
Effect of the ZnONPs and BZnO at different doses on the pup’s performance of male rats: (**A**) total, live, and dead litter numbers; (**B**) birth and weaning weight (g/dam); (**C**) survival rate (%); and (**D**) conception rate (%). All the values are expressed as the mean ± SEM. Different small letters (a–e) indicate significance at *p* < 0.0001. * *p* < 0.02 indicates a significant difference in ZnONP-2 and BZnO from other groups.

**Figure 5 antioxidants-10-00041-f005:**
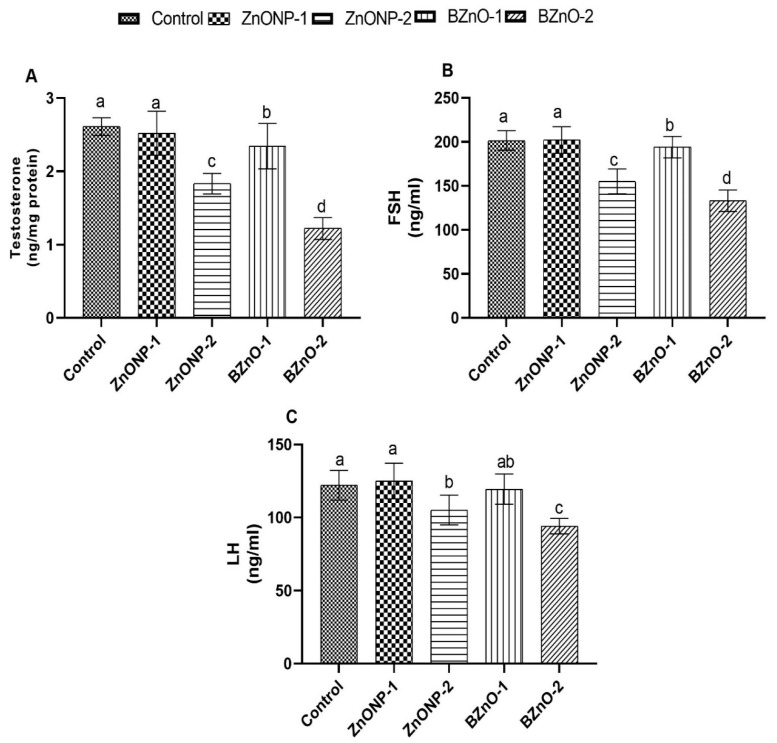
Effect of the ZnONPs and BZnO at different doses on the reproductive hormones of male rats: (**A**) testosterone level (ng/mg protein); (**B**) follicle-stimulating hormone (FSH) level (ng/mL); and (**C**) luteinizing hormone (LH) level (ng/mL). All the values are expressed as the mean ± SEM. Different small letters (a–c) indicate significance at *p* < 0.05.

**Figure 6 antioxidants-10-00041-f006:**
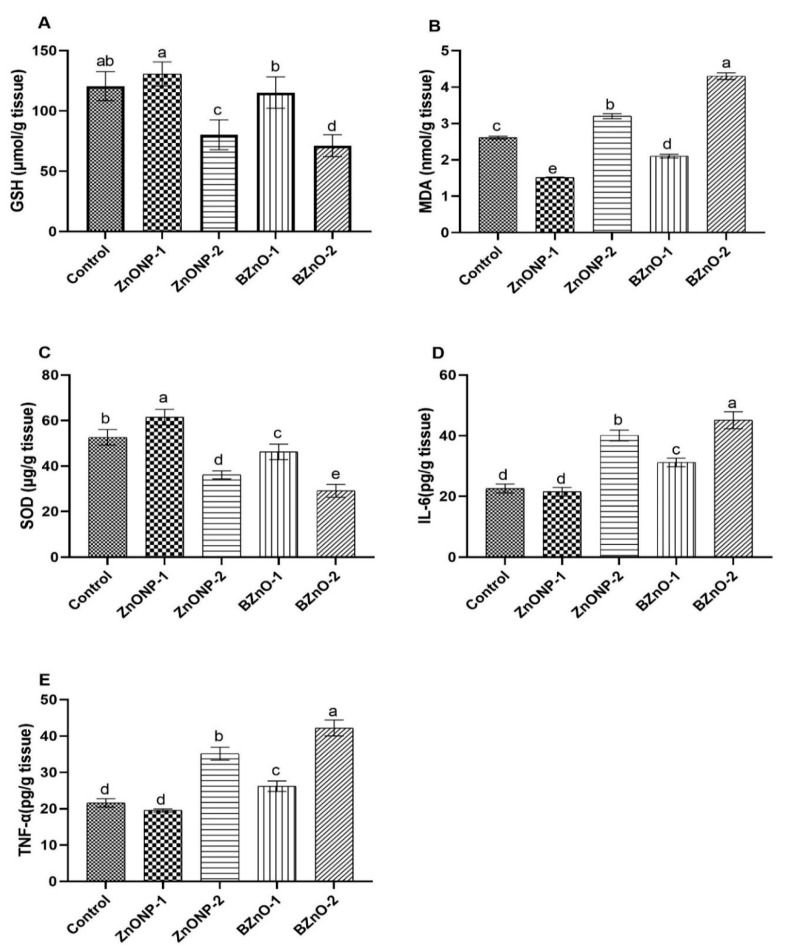
Effect of the ZnONPs and BZnO at different doses on the testicular oxidative markers: (**A**) glutathione (GSH) level (µmol/g tissue); (**B**) malonaldehyde (MDA) level (nmol/g tissue); (**C**) superoxide dismutase (SOD) level (µg/g tissue); (**D**) interleukin 6 (IL-6) level (pg/g tissue); and (**E**) tumor necrosis factor (TNF-α) level (pg/g tissue). All the values are expressed as the mean ± SEM. Different small letters (a–e) indicate significance at *p* < 0.05.

**Figure 7 antioxidants-10-00041-f007:**
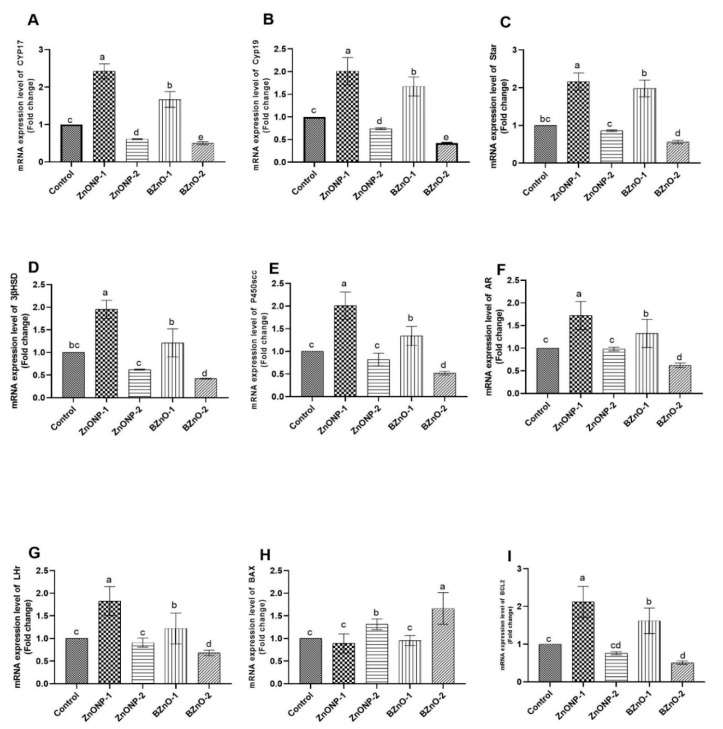
Effect of the ZnONPs and BZnO at different doses on testicular gene expression: (**A**) CYP17A1; (**B**) CYP19; (**C**) STAR; (**D**) 3β-HSD; (**E**) P450SCC; (**F**) AR; (**G**) LHR; (**H**) Bax, Bcl-2-associated X protein Bax; (**I**) Bcl-2, B-cell lymphoma 2 (Bcl2). All the values are expressed as the mean ± SEM. Different small letters indicate significance at *p* < 0.05.

**Figure 8 antioxidants-10-00041-f008:**
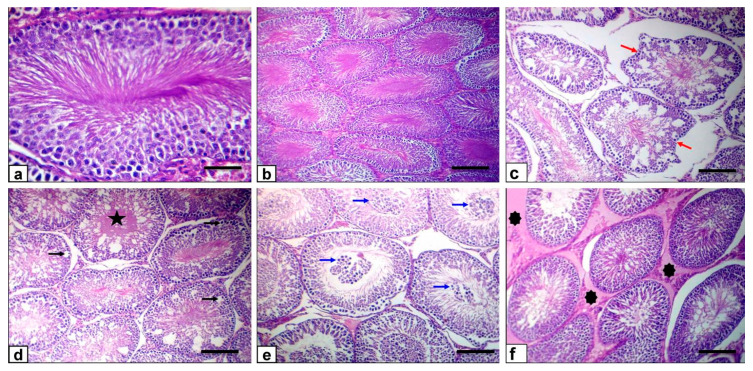
Photomicrograph of rat testes stained with HE: (**a**) the normal teste histoarchitecture of the control rats (scale bar = 50 µm); (**b**) ZnONP-1 rats showing normal seminiferous tubules with normal spermatogenesis (scale bar = 100 µm). (**c**–**f**) ZnONP-2 rats were showing shrunken, buckled, disorganized (red arrows) seminiferous tubules, depletion of germinal cells and hyalinization of the luminal contents (star), vacuolation (black arrows), and sloughing of the germinal epithelium in the lumen of the seminiferous tubules (blue arrows); this besides interstitial edema (asterisks), which is represented by the faint eosinophilic albuminous material (scale bars = 100 µm).

**Figure 9 antioxidants-10-00041-f009:**
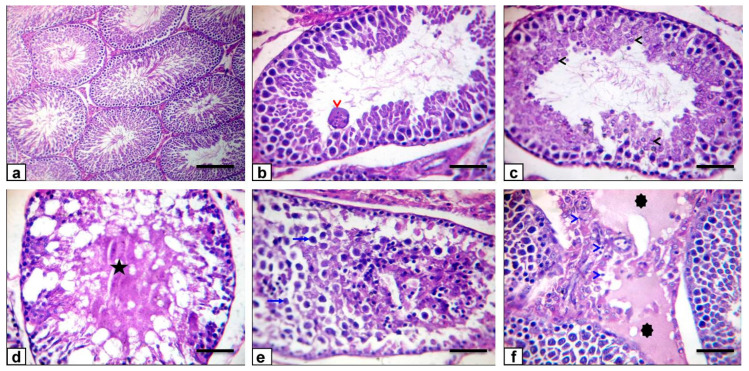
Photomicrograph of rat testes stained with HE: (**a**) BZnO-1 rats showing a normal histoarchitecture of the seminiferous tubules (scale bar = 100 µm). (**b**–**f**) BZnO-2 rats showing multi-nucleated giant cell formation in the lumen of some seminiferous tubules (red arrowhead), with fragmentation and necrosis of the germinal epithelium (black arrowheads); this besides necrotic seminiferous tubules, depletion of germinal cells and hyalinization of the luminal contents (star), sloughing of the germinal epithelium in the lumen of seminiferous tubules (blue arrows), and interstitial edema (asterisks), with mild inflammatory cell infiltration (blue arrowheads) (scale bars = 50 µm).

**Figure 10 antioxidants-10-00041-f010:**
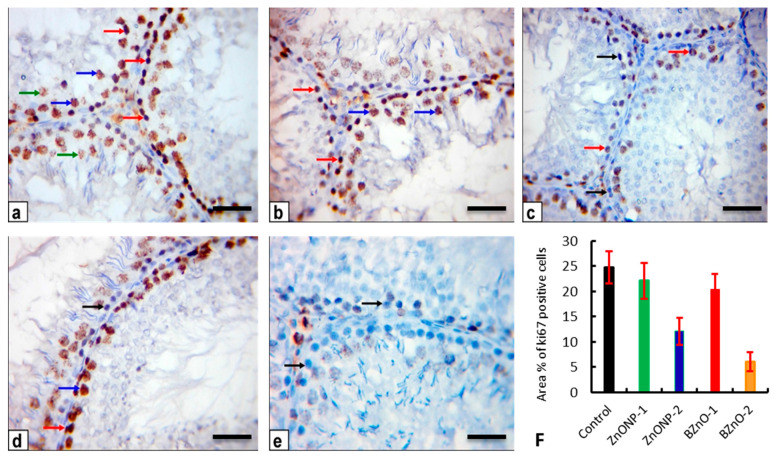
Photomicrograph of rat testes, showing positive brown Ki-67 immunostaining (scale bar = 50 μm): (**a**) control rats; (**b**) ZnONP-1 rats; (**c**) ZnONP-2 rats; (**d**) BZnO-1 rats; (**e**) BZnO-2 rats. Arrows in the panels are showing strong (red arrows), moderate (blue arrows), weak (green arrows), and negative (black arrows) immunoreactions in the nuclei of the spermatogonia and primary spermatocytes. (**f**) Area % of Ki-67 immunostaining positive cells. All the values are expressed as the mean ± standard error of the mean. Statistical significance was considered at a *p* < 0.0001.

**Table 1 antioxidants-10-00041-t001:** Primers for gene expression by RT-PCR.

Gene	Direction	Primer Sequence	Accession Number
***Bax***	Sense	GGCGAATTGGCGATGAACTG	NM_017059.2
	Antisense	ATGGTTCTGATCAGCTCGGG	
***Bcl-2***	Sense	GATTGTGGCCTTCTTTGAGT	NM_016993.1
	Antisense	ATAGTTCCACAAAGGCATCC	
***CYP17A1***	Sense	ACTGAGGGTATCGTGGATGC	NM_012753.2
	Antisense	TCGAACTTCTCCCTGCACTT	
***StAR***	Sense	CTGCTAGACCAGCCCATGGAC	NM_031558.3
	Antisense	TGATTTCCTTGACATTTGGGTTCC	
***Cyp11a1***	Sense	AGGTGTAGCTCAGGACTT	J05156
	Antisense	AGGAGGCTATAAAGGACACC	
***3β-HSD***	Sense	CCCATACAGCAAAAGGATGG	M38178
	Antisense	GCCGCAAGTATCATGACAGA	
***Cyp19***	Sense	GCTTCTCATCGCAGAGTATCCGG	M33986
	Antisense	CAAGGGTAAATTCATTGGGCTTGG	
**LHR**	Sense	CATTCAATGGGACGACTCTA	NM_012978.1
	Antisense	GCCTGCAATTTGGTGGA	
**AR**	Sense	TTTGGACAGTACCAGGGACC	NM_012502.1
	Antisense	CTTCTGTTTCCCTTCCGCAG	
***GAPDH***	Sense	TCAAGAAGGTGGTGAAGCAG	NM_017008.4
	Antisense	AGGTGGAAGAATGGGAGTTG	

Bax, Bcl-2-associated X protein. Bcl-2, B-cell lymphoma 2. CYP17A1, cytochrome P450 17A1. GAPDH, glyceraldehyde-3-phosphate dehydrogenase. STAR, steroidogenic acute regulatory protein. Cyp11a1, cholesterol side-chain cleavage enzyme mRNA (P450SCC) *3β-HSD*, 3-beta-hydroxysteroid dehydrogenase/delta-5-delta-4 isomerase type I; Cyp19, cytochrome P450 aromatase. AR, androgen receptors. LHR, luteinizing hormone receptors.

**Table 2 antioxidants-10-00041-t002:** Incidence and severity of the histopathological lesions in the testes of the ZnONP-2- and BZnO-2-treated rat groups.

Testes/Lesion	Incidence ^1^ and Severity ^2^ of Histopathological Lesions							
	ZnONP-2				BZnO-2			
	-	+	++	+++	-	+	++	+++
**Depletion of germinal cells**	3	2	1	0	2	1	2	1
**Hyalinization of the luminal contents**	4	1	1	0	2	1	2	1
**Vacuolation of germ cells and Sertoli cells**	3	1	2	0	2	2	2	0
**Sloughing of the germinal epithelium**	2	3	1	0	1	2	1	2
**Shrunken, buckled, disorganized**	2	2	2	0	2	3	1	0
**Interstitial edema**	2	1	2	1	1	1	2	2
**Giant cell formation**	6	0	0	0	2	3	1	0
**Interstitial inflammatory cell infiltration**	6	0	0	0	2	4	0	0

^1^ Number of rats with lesions per total examined (6 rats per group). ^2^ Severity of lesions was graded by estimating the percentage of area affected in the entire section. Lesion scoring: (-) absence of the lesion = 0%; (+) mild = 5–25%; (++) moderate = 26–50%; and (+++) severe =50% of the examined tissue sections.

## Data Availability

All data sets collected and analyzed during the current study are available from the corresponding author on reasonable request.

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
