# Peer review of "Insight Study on the Comparison between Zinc Oxide Nanoparticles and Its Bulk Impact on Reproductive Performance, Antioxidant Levels, Gene Expression, and Histopathology of Testes in Male Rats"

_antioxidants, 2020, doi:10.3390/antiox10010041_

Round 1

Reviewer 1 Report

THere are several issues in the manuscript which need to be rectified before any further decision.

The title is not clear. Testes of … ? Z O and N should not be capital in zinc oxide nanoparticles.

Z in zinc should not be capital.

The abstract is not structured, and authors are unable to establish the concept of the study.

Section 2.2: It is not clear why TEM was performed for commercial nanoparticles. Morphology of zinc oxide nanoparticles is well studied. The certificate of analysis from the supplier if good enough to establish the properties.

Section 2.3: The authors administrated equal volume (1 ml/rat) of test drugs. It is scientifically incorrect. The dose must be administrated based on the bodyweight of the rat. It is also not clear what is the concentration of the nanoparticles in the test preparations. Furthermore, 1 ml is quite a high volume; authors could make relatively concentrated solutions to reduce the volume of tween 80, as it has its toxicities.     

Section 3.7: The authors should provide a standard curve for delta delta CT. Authors are unable to explain the criteria for selecting different genes for the expression study.

Line 21: what type of beneficial effects?

Line 44: Is it sure that all nanoparticles have so-called “better functions”?  

Line 51: feed additive for what?

Line 79: what is fairly assigned?

Line 152: Is HSP70 studied?

Author Response

Comments and Suggestions for Authors

Reviewer 1

There are several issues in the manuscript which need to be rectified before any further decision.

  1. The title is not clear. Testes of ? Z O and N should not be capital in zinc oxide nanoparticles.

Response: Thank you for your comment. Title changed to: Insight study on the comparison between zinc oxide nanoparticles and its bulk impact on reproductive performance, antioxidant levels, gene expression, and histopathology of testes in male rats

  1. Z in zinc should not be capital. The abstract is not structured, and authors are unable to establish the concept of the study.

Response:  Thank you for your comment. We rewrite the abstract again considering your notes (please, see lines 21-41)

  1. Section 2.2: It is not clear why TEM was performed for commercial nanoparticles. Morphology of zinc oxide nanoparticles is well studied. The certificate of analysis from the supplier if good enough to establish the properties.

Response:  Thank you for your comment. TEM was used to make sure from the diameter of the nanoparticles used that it is the same as mentioned by the manufacture guide.  

  1. Section 2.3: The authors administrated equal volume (1 ml/rat) of test drugs. It is scientifically incorrect. The dose must be administrated based on the bodyweight of the rat. It is also not clear what is the concentration of the nanoparticles in the test preparations. Furthermore, 1 ml is quite a high volume; authors could make relatively concentrated solutions to reduce the volume of tween 80, as it has its toxicities.     

Response:  Thank you for your comment.

  • The doses was calculated according to body weight then dispersed in Tween 80 (10%) so the one ml contained the required dose and administered to each rat individually according to its weight. All drugs were dispersed in Tween 80 (10%) and each rat was injected intraperitoneally according to it weight to reach the desired dose, twice/week for eight consecutive weeks. (please, see line 84-86)
  • According to Varma RK, Kaushal R, Junnarkar AY, Thomas GP, Naidu MU, Singh PP, Tripathi RM, Shridhar DR. Polysorbate 80: a pharmacological study. Arzneimittelforschung. 1985;35(5):804-8. PMID: 4026903, the LD50 IP, rat (> 6804mg/kg) so one ml of Tween 80 was safe and nontoxic.
  1. Section 3.7: The authors should provide a standard curve for delta delta CT. Authors are unable to explain the criteria for selecting different genes for the expression study.

Response

  1. We tried to compare the results of Real-Time PCR with same samples in the case of using a standard curve, and without it, and had similar values. Of course, the use of a standard curve does not negate the need to use a housekeeping gene for data normalization. The disadvantage of using a standard curve is more consumption of reagents, so for some problems (no fund), we determine the relative gene expression values​​ without of using of standard dilutions.
  2. Then we calculate the efficiency of the each individual reaction using the data derived from the amplification curve and estimate the efficiency of amplification of each particular gene and substitute in the formula of delta delta Ct method the value of the efficiency.
  3. We use Relative quantification to compare gene expression levels across treatments we are interested only in relative quantification and compare all other samples to that using delta delta Ct 
  4. The primers used in this study was previously used in many studies. So, primer efficiencies in order to get the most accurate gene expression values as possible.
  5. Concerning the criteria for selecting the different genes are depend on the aim of our study in which The current study was investigating the Zinc oxide nanoparticle effect compared to its bulk form with different doses on male reproductive performance, testicular antioxidants, reproductive so we examine the gene expression for Apoptosis (Bax, Bcl2) and different genes related to the reproductive efficiency (STAR, steroidogenic acute regulatory protein. Cyp11a1, cholesterol side-chain cleavage enzyme mRNA (P450SCC) 3β-HSD, 3-beta-hydroxysteroid dehydrogenase/delta-5-delta-4 isomerase type I; Cyp19, cytochrome P450 aromatase. AR, Androgen receptors. LHR, Luteinizing hormone receptors).

  1. Line 21: what type of beneficial effects?

Response:  Thank you for your response. We added it (Please, see highlighted line 21-22)

  1. Line 44: Is it sure that all nanoparticles have so-called “better functions”?  

Response: Thank you for your comment. We corrected it (Please, see highlighted line 44-45).

  1. Line 51: feed additive for what?

Response: reedited to: The traditional Zinc oxide form had been used as a feed additive for animals feed (Please, see highlighted line 52)

  1. Line 79: what is fairly assigned?

Response:  Thank you for your comment we corrected it (Please, see highlighted line 80)

  1. Line 152: Is HSP70 studied?

Response. Thank you for your comment.  It is a typewriter unintended error. We deleted it

Reviewer 2 Report

Authors performed experiment to investigate the effects of zinc oxide nanoparticle on reproductive performance. ZnONP-1 injection improved sexual behavior, semen characteristics, and pup's performance. And, testicular antioxidants activity, GSH and SOD were increased. In this research, beneficial effects of zinc oxide nanoparticle on male reproduction. Therefore, these beneficial effects can be applied in another species and female reproduction. This manuscript has significance in nanoparticle application in male reproductive system. Scientific soundness and English writing is good. So, I recommend this manuscript as "Acceptance in present form"

Author Response

Reviewer 2

Authors performed experiment to investigate the effects of zinc oxide nanoparticle on reproductive performance. ZnONP-1 injection improved sexual behavior, semen characteristics, and pup's performance. And, testicular antioxidants activity, GSH and SOD were increased. In this research, beneficial effects of zinc oxide nanoparticle on male reproduction. Therefore, these beneficial effects can be applied in another species and female reproduction. This manuscript has significance in nanoparticle application in male reproductive system. Scientific soundness and English writing is good. So, I recommend this manuscript as "Acceptance in present form"

Response: Thank you for your comment

Reviewer 3 Report

The manuscript by Goma al, entitled “Insight Study on the impact of Zinc Oxide Nanoparticles on male reproductive performance, antioxidant levels, gene expression, and histopathology of testes in comparison to its bulk", describe the impact of zinc oxide nanoparticles, in comparison to its bulk form with different doses on male reproductive performance, testicular antioxidants, gene expression, and histopathology. In particular, the authors showed that zinc oxide nanoparticles 5 mg/kg, when compared to its bulk form, improved sexual behavior, semen characteristics, pup's performance as well as testicular antioxidants activity, GSH and SOD.

Although this study could be interesting, it has many shortcomings and inaccuracies. Moreover, some conclusions are not supported by experimental evidence. 

Major points:

  • Semen characteristics: the authors stated that sperm were retrieved from the epididymus (why not from the vas deferens?), but they didn’t indicate from what region. This is very important since sperm motility is gained during the passage into the epididymus and sperm from caput are immotile, while sperm from cauda display motility. In order to properly compare the experiments of this study, sperm must be retrieved from the same part of epididymus.
  • Gene expression analysis: the authors normalized gene expression analysis by using gapdh as a housekeeping gen. How this gene was selected? In the materials and methods section the authors don’t report if this gene has been validated as housekeeping gene in this tissue (under these treatments). The housekeeping validation is a prerequisite for gene expression analysis. For the housekeeping selection, please refer to Luddi et al., Fertility and Sterility, 2018.
  • Photomicrographs of rat testes are of bad quality. Moreover, a carefully stadiation of spermatogenesis in each tubule was not carried out. As the authors know, spermatogenesis is regulated by the hormones FSH and T in a stage-specific manner, e.g., androgens exert their effects on spermatogenesis primarily during stages VII–VIII, while FSH receptors are highest in Sertoli cells during stages XII–II. The damage induced by high dose of ZnONP as well as by its bulk must be better characterized in order to clearly define what kind of germinal cell is affected by this treatment.
  • A TUNEL analysis should be effective in demonstrating the type of tissue damage induced by Zn.
  • I recommend to the editor that the author(s) have their paper language edited

Author Response

Reviewer 3

The manuscript by Goma al, entitled “Insight Study on the impact of Zinc Oxide Nanoparticles on male reproductive performance, antioxidant levels, gene expression, and histopathology of testes in comparison to its bulk", describe the impact of zinc oxide nanoparticles, in comparison to its bulk form with different doses on male reproductive performance, testicular antioxidants, gene expression, and histopathology. In particular, the authors showed that zinc oxide nanoparticles 5 mg/kg, when compared to its bulk form, improved sexual behavior, semen characteristics, pup's performance as well as testicular antioxidants activity, GSH and SOD.

Although this study could be interesting, it has many shortcomings and inaccuracies. Moreover, some conclusions are not supported by experimental evidence. 

Major points:

  1. Semen characteristics: the authors stated that sperm were retrieved from the epididymus (why not from the vas deferens?), but they didn’t indicate from what region. This is very important since sperm motility is gained during the passage into the epididymus and sperm from caput are immotile, while sperm from cauda display motility. In order to properly compare the experiments of this study, sperm must be retrieved from the same part of epididymus.

Response: Thank you for your response.

Thank you for your response. We retrieved sperm from the same part in which cauda epididymis was isolated, minced, and suspended in 15 mL of a Biggers-Whitten-Whittingham (BWW) medium. BWW is a medium used essentially to sustain the fertilizing potential of spermatozoa in vitro, supplemented with, bovine serum albumin (BSA), glucose, pyruvate, and lactate, as well as Ca2+ and HCO3 (Sigma – Aldrich, Saint Louis, Missouri, USA)

  1. Gene expression analysis: the authors normalized gene expression analysis by using gapdh as a housekeeping gen. How this gene was selected? In the materials and methods section the authors don’t report if this gene has been validated as housekeeping gene in this tissue (under these treatments). The housekeeping validation is a prerequisite for gene expression analysis. For the housekeeping selection, please refer to Luddi et al., Fertility and Sterility, 2018.

Response:

Thank you for your comment. We selected the GAPDH depend on many previous studies  such as, (Barlow et al. 2003; Lee et al. 2004; Bhushan et al. 2008; Nemoto et al. 2009; Dutta et al. 2017; Hirai et al. 2017; Ning et al. 2018a; Ning et al. 2018b; Schaalan et al. 2018; Zhang et al. 2020) .Although, total RNA from testes was separated into four aliquots for reverse transcription (RT), with one aliquot receiving no enzyme and designated to serve as a negative control. Quality of RT reactions was confirmed by comparison of triplicate RT versus no enzyme control for each RNA sample using the glyceraldehyde-3-phosphate dehydrogenase (GAPDH) primer set.

  1. Photomicrographs of rat testes are of bad quality. Moreover, a carefully stadiation of spermatogenesis in each tubule was not carried out. As the authors know, spermatogenesis is regulated by the hormones FSH and T in a stage-specific manner, e.g., androgens exert their effects on spermatogenesis primarily during stages VII–VIII, while FSH receptors are highest in Sertoli cells during stages XII–II. The damage induced by high dose of ZnONP as well as by its bulk must be better characterized in order to clearly define what kind of germinal cell is affected by this treatment. A TUNEL analysis should be effective in demonstrating the type of tissue damage induced by Zn.

I recommend to the editor that the author(s) have their paper language edited

Response:

Thank you for your comment.

  • The quality of photomicrographs were adjusted, (please, see the figure). In addition, table 2 showed the Incidence and severity of histopathological lesions in the testes of ZnONP-2 and BZnO-2 treated rat groups. For apoptosis we use Real-time Polymerase Chain Reaction (RT-PCR) for assessed the apoptotic gene expression as well as, using Immunohistochemistry and Quantitative Analysis The ki-67 expression used to detect the proliferative index in different treated rats.
  • Assessment of Spermatogenesis indexes were done using Johnsen scores to sort spermatogenesis at 400× magnification power. To evaluate the quality of the specimen, we used a slightly modified Johnsen score count (Johnsen 1970). Microscopically, the spermatogenesis values were classified as 1, for Sertoli cells only; 2, for spermatogonia only; 3 and 4, for no further than primary spermatocytes (40 spermatids per view, respectively); 8, 9, and 10, for maturation phase spermatids (50 spermatids per view, respectively). (Please, see lines 168-173, 239,240,435,).
  • Additionally, We take all of your valuable comments into consideration in the process of continuing this work ( about A TUNEL analysis, and uses special stains as will as electron microscope analysis to determine spermatogenesis stages

  • The whole manuscript was revised by a native English speaker

Barlow, NJ, Phillips, SL, Wallace, DG, Sar, M, Gaido, KW, Foster, PM (2003) Quantitative changes in gene expression in fetal rat testes following exposure to di (n-butyl) phthalate. Toxicological sciences 73, 431-441.

Bhushan, S, Tchatalbachev, S, Klug, J, Fijak, M, Pineau, C, Chakraborty, T, Meinhardt, A (2008) Uropathogenic Escherichia coli block MyD88-dependent and activate MyD88-independent signaling pathways in rat testicular cells. The Journal of Immunology 180, 5537-5547.

Dutta, D, Park, I, Guililat, H, Sang, S, Talapatra, A, Singhal, B, Mills, NC (2017) Testosterone regulates granzyme K expression in rat testes. Endocrine Regulations 51, 193-204.

Hirai, S, Hatayama, N, Naito, M, Nagahori, K, Kawata, S, Hayashi, S, Qu, N, Terayama, H, Shoji, S, Itoh, M (2017) Pathological effect of arterial ischaemia and venous congestion on rat testes. Scientific reports 7, 1-9.

Johnsen, SG (1970) Testicular biopsy score count--a method for registration of spermatogenesis in human testes: normal values and results in 335 hypogonadal males. Hormones 1, 2-25.

Lee, K-F, Yeung, WS, Chow, JF, Shum, CK, Luk, JM (2004) Different testicular gene expression patterns in the first spermatogenic cycle of postnatal and vitamin A-deficient rat testis. Biology of reproduction 70, 1010-1017.

Nemoto, K, Miyajima, S, Hara, S, Saigusa, R, Yamada, M, Shikama, H, Yotsuya, S, Sekimoto, M, Degawa, M (2009) Decreased Gene Expression of Testicular Cell-Specific Proteins in Cadmium-Induced Acute Testicular Toxicity. Journal of health science 55, 952-956.

Ning, J-Z, Li, W, Cheng, F, Rao, T, Yu, W-M, Ruan, Y, Yuan, R, Zhu, S-M, Zhang, X-B, Du, Y (2018a) The protective effects of GYY4137 on testicular torsion/detorsion injury in rats. Int. J. Clin. Exp. Med 11, 3387-3395.

Ning, JZ, Li, W, Cheng, F, Rao, T, Yu, WM, Ruan, Y, Yuan, R, Zhang, XB, Du, Y, Xiao, CC (2018b) The protective effects of GYY4137 on ipsilateral testicular injury in experimentally varicocele‑induced rats. Experimental and therapeutic medicine 15, 433-439.

Schaalan, MF, Ramadan, BK, Abd Elwahab, AH (2018) Ameliorative effect of taurine-chloramine in azathioprine-induced testicular damage; a deeper insight into the mechanism of protection. BMC complementary and alternative medicine 18, 1-14.

Zhang, T-D, Ma, Y-B, Li, H-C, Chong, T, Wang, Z-M, Zhang, L-D (2020) Low Dose of Genistein Alleviates Mono-(2-Ethylhexyl) Phthalate-Induced Fetal Testis Disorder Based on Organ Culture Model. Oxidative Medicine and Cellular Longevity 2020,

Round 2

Reviewer 1 Report

The authors addressed the points and now it is ready for publication. The authors are advised to proofread once the units/values associated with the study. 

Reviewer 3 Report

The Authors improved the first  submission of the manuscript even if some requested of the reviewer were not agree, for example the validation of the housekeeping gene, the stadiation of the spermatogenetic process in order to evidence the exact moment of the defect sunrise and the use of the Tunel method in order to highlight the apoptosis in the germ cell epithelium.